# Checkpoint Inhibitors in Dogs: Are We There Yet?

**DOI:** 10.3390/cancers16112003

**Published:** 2024-05-24

**Authors:** Antonio Giuliano, Pedro A. B. Pimentel, Rodrigo S. Horta

**Affiliations:** 1Department of Veterinary Clinical Science, Jockey Club College of Veterinary Medicine, City University of Hong Kong, Hong Kong, China; 2Veterinary Medical Centre, City University of Hong Kong, Hong Kong, China; 3Department of Veterinary Medicine and Surgery, Veterinary School, Universidade Federal de Minas Gerais (UFMG), Belo Horizonte 31270-901, MG, Brazil; pedrobpimentel@gmail.com

**Keywords:** PD-1, PD-L1, CTLA-4, canine cancer, monoclonal antibodies, immunotherapy

## Abstract

**Simple Summary:**

Immune checkpoints are essential to the body’s reaction to immunological stimuli. The most studied immune checkpoint molecules are programmed death (PD-1) with its ligand (PD-L1) and cytotoxic T-lymphocyte-associated protein 4 (CTLA-4) with its ligands CD80 (B7-1) and CD86 (B7-2). Certain tumours can evade immunosurveillance by activating these immunological checkpoint targets. These proteins are often upregulated in cancer cells and tumour-infiltrating lymphocytes, allowing cancer cells to escape immune surveillance and promote tumour growth. By blocking inhibitory checkpoints, immune checkpoint inhibitors (ICI) can help restore the immune system to fight cancer effectively. Emerging studies in veterinary oncology are centred around developing and validating canine-targeted antibodies. Anti-PD-1 and anti-PD-L1 treatments emerge as particularly promising among ICIs, reflecting the notable achievements in human medicine over the last decade. This article aims to review the current research literature about the expression of immune checkpoints in canine cancer and the current results of ICI treatment in dogs.

**Abstract:**

Immune checkpoint inhibitors (ICI) have revolutionised cancer treatment in people. Immune checkpoints are important regulators of the body’s reaction to immunological stimuli. The most studied immune checkpoint molecules are programmed death (PD-1) with its ligand (PD-L1) and cytotoxic T-lymphocyte-associated protein 4 (CTLA-4) with its ligands CD80 (B7-1) and CD86 (B7-2). Certain tumours can evade immunosurveillance by activating these immunological checkpoint targets. These proteins are often upregulated in cancer cells and tumour-infiltrating lymphocytes, allowing cancer cells to evade immune surveillance and promote tumour growth. By blocking inhibitory checkpoints, ICI can help restore the immune system to effectively fight cancer. Several studies have investigated the expression of these and other immune checkpoints in human cancers and have shown their potential as therapeutic targets. In recent years, there has been growing interest in studying the expression of immune checkpoints in dogs with cancer, and a few small clinical trials with ICI have already been performed on these species. Emerging studies in veterinary oncology are centred around developing and validating canine-targeted antibodies. Among ICIs, anti-PD-1 and anti-PD-L1 treatments stand out as the most promising, mirroring the success in human medicine over the past decade. Nevertheless, the efficacy of caninized antibodies remains suboptimal, especially for canine oral melanoma. To enhance the utilisation of ICIs, the identification of predictive biomarkers for treatment response and the thorough screening of individual tumours are crucial. Such endeavours hold promise for advancing personalised medicine within veterinary practice, thereby improving treatment outcomes. This article aims to review the current research literature about the expression of immune checkpoints in canine cancer and the current results of ICI treatment in dogs.

## 1. Introduction

Immune checkpoint inhibitors (ICI) have revolutionised cancer treatment, offering new hope for patients with advanced melanoma and other solid tumours [1,2]. The application of checkpoint inhibitors in veterinary oncology, particularly in dogs with cancer, has also emerged as an area of active research, potentially transforming the treatment landscape for canine malignancies [3,4,5,6].

The immune system plays an important role in cancer surveillance. Failure of cancer surveillance is responsible for cancer development and progression, and cancer uses different tactics to evade the immune system [7]. An increased number and activation of immune suppressive cells like tumour-associated macrophages, regulatory T-cells, and myeloid-derived cells are responsible for dampening the immune response to cancer. In addition, tumours release various compounds like nitric oxide, reactive oxygen species (ROS), and immunosuppressive cytokines including TGF-β and IL-10 that inhibit the immune system [7,8,9]. Reduced expression of major histocompatibility molecules (MHC) and an increase in cancer metabolic products like indoleamine 2,3-dioxygenase (IDO) also contribute to cancer immune evasion [9]. Another very important mechanism of cancer immune escape is the upregulation of immune checkpoint proteins. The most studied checkpoint proteins are PD-1/PDL1 and CTLA-4, but more recently, LAG-3, TIM-3, CD200, and OX-40 have also been investigated [10,11,12]. The functions of these proteins and their potential roles in canine and human cancer will be reviewed. Furthermore, all the available literature on the expression of immune checkpoints in canine cancer and the current results of ICI treatment in dogs will be summarised and discussed.

## 2. Role of Check Point Proteins in Human Cancer and Available ICI

PD-1 is a co-inhibitory receptor expressed on the surface of T cells, and it is crucial in regulating T cell exhaustion. T cell activation is suppressed, and downstream signalling pathways are triggered when PD-1 interacts with its ligands, PD-L1 and PD-L2 [10,13] (Figure 1).

The expression patterns of PD-L1 and PD-L2 are different. Hematopoietic cells (like T cells, NK cells, dendritic cells, macrophages, and B cells), as well as non-hematopoietic cells (like vascular and lymphatic endothelial cells and epithelial cells), have high expression of PD-L1. PD-L2 is only expressed in hematopoietic cells, especially B and T cells [10,13]. High levels of PD-L1 expression in tumour cells and antigen-presenting cells in the tumour microenvironment are one of the main mechanisms by which tumour immune evasion occurs [10,13]. In certain tumour types, like non-small cell lung cancer (NSCLC) and melanoma, the expression of PD-L1 inside the tumour is linked to improved responses to PD-1/PD-L1 checkpoint blockage [14]. However, the significance of high PD-L1 expression without infiltrating T-lymphocytes is questionable and could explain why some patients with high PD-L1 expression do not respond to ICI targeting PD/PD-L1 [15].

There are three FDA-approved fully humanised IgG4 monoclonal antibody (mAb) checkpoint inhibitors against PD-1 in people, i.e., pembrolizumab, nivolumab, and cemiplimab [16]. Pembrolizumab and nivolumab are approved for the treatment of metastatic melanoma and NSCLC, while cemiplimab was approved for advanced-stage cutaneous squamous cell carcinoma [17,18,19,20,21,22,23,24]. Interestingly, independently from the tumour type, pembrolizumab has been used for the treatment of advanced solid tumours with a high mutational burden, suggesting that a high mutational burden could be another predictor of response to ICI [21]. A large, randomised phase 3 trial proved that pembrolizumab appeared more effective in advanced melanoma compared to ipilimumab [25]. However, the combination of nivolumab and ipilimumab appeared superior as a first-line treatment to either agent alone in metastatic melanoma in people [26,27].

Atezolizumab is a humanised IgG1 anti-PD-L1 mAb that was originally approved for the treatment of urothelial carcinoma and later on also approved for NSCLC, melanoma, and hepatocellular carcinoma [28,29,30,31]. Other PD-L1 checkpoint inhibitors are Durvalumab approved for similar indications as atezolizumab and avelumab approved for Merkel cell carcinoma, but also urothelial and renal cell carcinoma [16].

Ipilimumab was the first and only approved human IgG1 mAb against CTLA-4, and it was originally approved for metastatic melanoma [32,33].

CTLA-4 is another checkpoint receptor, primarily expressed in T cells, with limited expression observed in B cells [34,35]. The interaction of CTLA-4 with its ligands, CD80 (B7-1) and CD86 (B7-2) receptors, expressed in various immune cells and especially antigen-presenting cells, is responsible for early T-cell reduced activation [10,36]. Following antigen recognition (T cell receptor and MHC interaction), the binding of CD28 with CD80 and CD86 increases T-cell activation [37]. The ligands for CTLA-4 and CD28 are identical, and the binding of CTLA-4 on T cells competes with CD28 for the binding of their ligands, CD80 and CD86. When CTLA-4 binds to its ligand, it reduces the activation of the cytotoxic T-cells and suppresses the immune response to tumours [16,36]. Tregs infiltrating the tumour constitutively express CTLA-4, contributing to creating an immunosuppressive environment in cancer [38] (Figure 1).

Another more recently discovered checkpoint is the lymphocyte-activation gene 3, known as LAG3. T-lymphocytes display this protein on their surface, especially when exhausted or dysfunctional [39,40,41]. LAG3 binds to MHC class II molecules on the surface of antigen-presenting cells, inhibiting both CD4 and CD8 T-cell activation and proliferation [39,40]. LAG-3 also inhibits CD8 T-cell function by binding to a second ligand called galectin-3 that is expressed in activated T lymphocytes but also in a variety of human cancers [39,42]. Currently, a few clinical trials are investigating LAG-3 inhibitors in people, and recently, the LAG-3 inhibitor relatlimab has been approved in combination with nivolumab for patients with metastatic melanoma [42,43]. Although comparison within trials is challenging, there was comparable efficacy between relatlimab and nivolumab when compared to ipilimumab and nivolumab, but adverse events seem less common with relatlimab and nivolumab [27,43,44].

T-cell immunoglobulin mucin-3 (TIM-3) is a recently described immune checkpoint molecule that is emerging as a promising target for cancer immunotherapy. TIM-3 is a negative regulatory immune checkpoint found in T cells, including regulatory T-cells, but also in B-cells, dendritic cells, and macrophages [45,46]. It has multiple ligands, such as galectin-9 (Gal-9), phosphatidylserine, high mobility group protein 1 (HMGB1), and carcinoembryonic antigen cell adhesion molecule 1 (CEACAM-1) [47].

TIM-3 and Gal-9 have been found to be expressed in a variety of cancers, and the mechanism of immune evasion is likely related to the binding of TIM-3 to Gal-9, which induces T-cell apoptosis and increased expression of TIM-3 on regulatory T-cells, promoting T-cell exhaustion [48]. Tim-3 can also regulate the conversion from a pro-inflammatory M1 macrophage to an anti-inflammatory and tumour-permissive M2 macrophage [48,49].

VISTA (V-domain Ig-containing suppressor of T cell activation) is another newly discovered checkpoint. VISTA is expressed by T cells and myeloid cells [50]. VISTA exhibits a strong affinity for Gal-9 and amplifies the immunosuppressive properties of TIM-3. When a multiprotein complex made up of VISTA, Gal-9, and TIM-3 assembles on the surface of T cells, the granzyme B contained in CD8 T-cells is kept from leaving the cells. Granzyme B activation inside the cell results in mitochondrial malfunction, apoptosis, and T-cell-programmed death [48,50]. A few trials have investigated the potential benefit of TIM-3 inhibitors in people with various malignancies, and sabatolibmab has shown some promising results in myelodysplastic syndrome and chronic myeloblastic leukaemia [51].

Various types of immune cells express another recently discovered co-stimulatory molecule, the tumour necrosis factor receptor superfamily member 4 (OX40) (CD134). OX40 binds to its ligands (OX40L) in APC, which activates T-cells, reduces the immunosuppression of regulatory T (Treg) cells, and increases the immune response [52,53]. Agonistic monoclonal antibodies activating the costimulatory molecules OX40/OX40L are likely to be effective in cancer treatment [52,53]. At the time of writing, no OX40 antibodies have been approved by the FDA or any other regulatory bodies. However, there are several OX40 antibodies in clinical trials that have shown potential benefits in advanced solid tumours [52,53].

CD200, also known as OX-2 membrane glycoprotein, is a transmembrane protein predominantly expressed in T cells. Upon interaction with its corresponding receptor, CD200R, located on antigen-presenting cells, it delivers immunosuppressive signals [12]. Notably, CD200 is frequently expressed in people with chronic lymphocytic leukaemia (CLL) and serves as a diagnostic marker. Elevated serum levels of CD200 have been correlated with poorer prognostic outcomes in CLL patients [54]. Furthermore, investigations have demonstrated promising outcomes of anti-CD200 therapy in the treatment of gliomas in dogs and also hold potential as a treatment modality for CLL in dogs [55].

## 3. The Expression and Clinical Relevance of Immune Checkpoints in Dogs with Cancer

### 3.1. PD/PD-L1 in Dogs

The PD/PD-L1 expression in cancer has been investigated not only in humans but also in dogs. In physiologic conditions, the expression of PD-1 in dogs’ T cells varies around 5–10% of CD4^+^ and 20–25% of CD8+, and like in humans, it is an essential mechanism of immune response and regulation [56]. PD-1 expression in T-cells can become upregulated in different malignant tumours in dogs, promoting tumour immune escape [56,57].

Several methods can be used to evaluate PD-L1 expression. Immunohistochemistry is currently considered the gold standard for selecting human cancer patients eligible for anti-PD-1 and anti-PD-L1 immunotherapy [58]. However, technical aspects such as the sample type, the antibody used, and the reference cutoffs may provide discordant results [59]. Validation of commercial antibodies designed for human samples but with proven cross-reactivity for canine tissues may increase their availability for routine testing. While there are no specifically established evaluation methods for canine tumours, the extrapolation of human criteria has been used by Maekawa et al. (2021) and Muscatello et al. (2023) in several canine neoplasms [4,59]. In these studies, the PD-1/PD-L1 tumour proportion score (TPS; percentage of positive neoplastic cells) and combined positive score (CPS; percentage of positive cells, including immune cells, among total neoplastic cells) have been investigated [4,59]. An alternative method, the immune cell density score (IDS), which considers the proportion of positive immune cells among all the cells in a specific area, has been reported [60]. These scores may be used for PD-1 and PD-L1 analysis in dogs [4,59,60].

While the validation of anti-canine PD axis checkpoint molecule antibodies is still under study, the expression of mRNA for these clusters of differentiation through real-time quantitative polymerase chain reaction (RT-qPCR) and in situ hybridization provides viable alternatives [60,61].

Expression of PD-L1 was evaluated in several canine tumours (Table 1). PD-L1 immunolabeling was consistently high in canine melanoma, as demonstrated in 95% of oral melanomas (19/20), with most presenting TPS higher than 49% (18/20) [4] and 100% among 17 melanomas (13 oral and 4 cutaneous), with the TPS ranging from 1–14% [59]. Among 17 squamous cell carcinomas (13 cutaneous and 4 oral), there was only 18% (3/17) of positivity, with a TPS of 17–64% and a CPS of 17–78% [59], but Maekawa et al. (2021) found 90% of immunolabeling among 20 cutaneous squamous cell carcinomas (18/20), with a TPS higher than 49% in 80% of cases (16/20).

Among canine mammary gland carcinoma cases, positivity for PD-L1 ranged from 3–100% in two studies [57,58]. Such differences might be related to the heterogeneity of histological and molecular subtypes. In women, PD-L1 expression in breast cancer is associated with negative hormone receptors, HER2 overexpression, and triple-negative PTEN-mutated carcinomas [62], but such an association has not yet been identified in dogs. Also, differences may be attributed to the antibody used or the evaluation process. It can be challenging to eliminate cytoplasmic background, which impairs the assessment of membrane labelling, and only cellular localization should be considered [59,63]. PD-1 is largely expressed on both CD8^+^ (70.9–96.6%) and CD4^+^ (80.2–96.8%) tumour-infiltrated lymphocytes (TILs) obtained from canine oral melanoma, demonstrating upregulation of PD-1 possibly leading to functional exhaustion of TILs in this cancer type [55]. Furthermore, CD3^+^ T cells are increased in canine melanoma with higher PD-1 tumour expression, whereas the expression of IBA1+ macrophages and CD79a^+^ B cells was not associated with changes in the expression of the PD axis (PD-1, PD-L1, and PD-L2) [61]. This suggests a modulation of T cells rather than macrophages in canine melanoma.

Immunohistochemistry, western blot, immunofluorescence, and mRNA (RT-PCR) have been used to investigate the expression of PD-L1 in urothelial carcinoma in dogs [64,65]. High expression of PD-L1 has been consistent in those studies, suggesting that canine urothelial carcinoma could be a good translational model of urothelial carcinoma in people. This tumour, compared to other solid tumours, is relatively more responsive to immunomodulators and immunotherapy in people [66,67]. Similarly to people, other immunotherapy treatments like Bacillus Calmette–Guérin (BCG) and nano-immunotherapy have proven to affect PD-1/PD-L1 expression in canine urothelial carcinoma [65].

Canine-diffuse large B-cell lymphoma (DLBCL) often expresses a high level of PD-L1 measured by in situ hybridization (semi-quantitative analysis), and high expression has been associated with a worse prognosis than cases with a low score [60]. However, in the same study, CD8^+^ expression in tumour-infiltrating lymphocytes (TILs) was not associated with PD-L1 or PD-1 expression. Samples analysed by flow cytometry from peripheral blood and lymph nodes in dogs with B-cell high-grade lymphoma showed higher PD-1 expression in CD4^+^ and CD8^+^ T cells than those in clinically healthy dogs. Elevated levels of CD4^+^ PD-1+ lymphocytes were associated with substage B lymphoma but not with survival time [68]. However, higher values of CD4^+^ PD-1+ in lymph nodes were associated with a shorter survival time with a cutoff of 75.9%, resulting in median survival times of 135 days versus only 13 days [68]. The expression of PD-1 in TILs CD4+ appears to play a more significant role as a prognostic marker than TILs CD8+, serum CD8+, and serum CD4+ [60,68]. The expression of PD-L1 in malignant mammary gland tumours in dogs is notably higher compared to benign mammary gland tumours [69]. Higher immunostaining scores of PD-L1 were associated with shorter overall survival times, using a cutoff of 188/HPF. The employed scoring system was calculated by multiplying the staining intensity (1–4) by the percentage of stained tumour cells. Furthermore, using immunohistochemistry, the expression of the PD axis in these cancers’ cells seems more reliable in prognostication than its expression in tumour-infiltrating immune cells, identified in only 26.8% (11/41) of cases and not associated with survival time [69]. However, the immunohistochemistry score systems evaluating staining intensity could lack reproducibility and be subject to pathologist interpretation, jeopardising, the comparison between the results of veterinary studies. Mirroring human medicine standardisation and validation of these tests using specific staining platforms should be implemented in veterinary medicine.

In dogs with apocrine gland anal sac adenocarcinoma treated with surgery alone, PD-L1 expression in IHC was associated with survival time: dogs with PD-L1+ tumours presented a median survival time of 235 days versus 576 days in dogs with PD-L1 tumours (*p* = 0.022) [70]. PD-1 and PD-L1 might also be assessed through liquid biopsy, as demonstrated by Song et al. (2021) in dogs with various neoplasms. Higher levels of circulating PD-1 and PD-L1 were demonstrated in dogs with different tumours in comparison to healthy dogs, which indicates it could be a potential biomarker for diagnosis and therapy monitoring [70].

### 3.2. CTLA-4

The CTLA-4 protein expression in CD4^+^ and CD8^+^ T-cells has been fully investigated in people but poorly in dogs [68,71]. In canine oncology, CTLA-4 studies are restricted to B-cell high-grade lymphoma, mammary gland, histiocytic, and melanocytic tumours [68,69,71,72]. The process of CTLA-4 and CD28 costimulatory capture of ligands (CD80 and CD86) from neighbouring cells was identified in humans and has also been described in Tasmanian devils [73]. This demonstrates the stability of this immune mechanism throughout mammalian evolution.

The membrane form of CTLA-4 (mCTLA-4) and the serum-soluble CTLA-4 (sCTLA-4) constitute distinct isoforms. mCTLA-4 is frequently upregulated in tumours, rendering it a prominent subject of investigation in CTLA-4 studies and correlating it with aggressive cancer [69]. Conversely, sCTLA-4 levels typically show an elevation in autoimmune disorders and cancers [74,75]. Hence, careful consideration of the specific isoform of CTLA-4 is warranted, as it can potentially introduce bias in research outcomes and evoke varied responses.

On flow cytometry, the expression of CTLA-4 in peripheral CD4^+^ T cells was found to be elevated in dogs with B-cell multicentric high-grade lymphoma compared to the clinically healthy control group [68]. Dogs with higher CD4^+^ CTLA-4^+^ levels also exhibited a significantly lower median survival time, decreasing from 138 days to 11 days, corresponding to the groups with higher and lower CD4^+^ CTLA-4^+^ levels (*p* < 0.001).

Cytoplasmic immunostaining of CTLA-4 is positively associated with shorter specific survival times in canine mammary gland tumours [69], using an IHC cutoff score of 177/HPF. The score employed was semi-quantitative (the intensity of staining multiplied by the % of stained cells). This could suggest a role for CTLA-4 in tumour progression, possibly due to T-reg control over immune local response.

In canine melanocytic tumours, higher immunostaining expression of CTLA-4 was identified in oral melanoma compared to benign cutaneous melanomas, and it was associated with a worse prognosis using a cutoff of 2.2 cells/HPF [72].

Using flow cytometry, the expression of CTLA-4 was measured in CD4^+^ and CD8^+^ T cells in the peripheral blood of dogs with histiocytic sarcoma [71]. The results showed a higher expression of CTLA-4 compared to healthy dogs. PD-1 expression in serum CD8+ T cells was also higher in dogs with histiocytic sarcoma than in control dogs [71].

No studies have yet reported CTLA-4 expression in canine transmissible venereal tumours (CTVT). However, the stability of CTLA-4/CD28 capture of ligands in mammals may help to comprehend the immune evasion of transmissible devil facial tumours and transmissible venereal tumours [73]. While the loss of MHC and decrease of NK activity are fundamental for CTVT growth [76], further studies comparing different phases of the tumour’s growth, CTLA-4 expression in the tumour and local immune cells, and gene expression of CTLA-4 may provide more reliable evidence of CTVT immune escape.

CTLA-4 expression in B-cells is not well understood in dogs. While known for regulating T-cell function, CTLA-4 also contributes to immune tolerance in B-cells. The lack of CTLA-4 in B-cells can lead to increased autoantibody production through epigenetic and transcriptional pathways [35]. However, specific mechanisms underlying CTLA-4 expression in B-cells within the tumour microenvironment are uncharacterised. Further research is required to determine whether B-cell-mediated immune responses in tumourigenesis and tumour progression are influenced by CTLA-4 expression in dogs. Table 2 shows data on the main tests employed for measuring CTLA-4 expression in canine tumours.

### 3.3. Clinical Trials of Immune Checkpoint Inhibitors in Dogs

The advent of clinical trials with ICIs in veterinary medicine intensified in this decade, preceded by the growth of research in people and approval of the first ICIs since 2011: ipilimumab (anti-CTLA-4), pembrolizumab (anti-PD-1), and nivolumab (anti-PD-1) [77,78]. Therapies targeting the PD-1/PDL-1 and CTLA-4 checkpoints were pioneers in human medicine, and a few studies have investigated a similar therapeutic approach in canine tumours [6,77,79,80].

The first pilot study of ICI in dogs was performed in Japan, where the in vitro immunomodulatory effects of c4G12, a canine-chimerised anti-PD-L1 monoclonal antibody, and the clinical efficacy in dogs with various cancers were evaluated [5]. The c4G12 demonstrated enhanced in vitro immunostimulation with increased cytokine production and proliferation of dog peripheral blood mononuclear cells. The safety profile and clinical efficacy of c4G12 were evaluated in seven dogs with advanced melanoma (stages from II to IV) and two dogs with undifferentiated sarcoma with multiple metastases. The dogs were treated with 3 and 5 mg/kg of c4G12 intravenously every 2 weeks, and objective response, including complete and partial response, was achieved in one of seven dogs with melanoma and in one of two dogs with undifferentiated sarcoma with multiple distant metastases. Interestingly, the only dog with melanoma that responded had a localised stage II disease, and response was achieved only after 10 weeks (after five injections). Even more interestingly, the cancer progressed later on, and the dose was increased to 5 mg/kg at week 24. Again, a response of around 80% was achieved only at 34 weeks. The other dog with undifferentiated sarcoma also achieved a partial response (PR) with a 34% reduction in size. The response was achieved after 3 weeks of treatment at 5 mg/kg. The treatment was very well tolerated, with only one patient developing grade 1 diarrhoea.

A couple of years later, Igase et al. (2020) investigated the use of both a rat–dog chimeric and caninized anti-canine PD-1 monoclonal antibodies. Investigations of in vitro functionality and immunostimulation were combined with a small pilot clinical study to investigate the safety and efficacy of various tumours [79]. Both antibodies showed binding affinity to PD-1 and an inhibitory effect on the PD-1/PD-L1 interaction, with increased but variable interpatient levels of IFN-γ production. In the same study, the authors evaluated the monoclonal antibodies in 30 patients with various malignancies. Of the 30 cases treated, 21 were advanced-stage oral malignant melanomas (OMM) (four cases of Stage III and 17 cases of Stage IV). The remaining tumours included three cutaneous melanomas and one of each mammary gland tumour (MGT), squamous cell carcinoma (SCC), renal carcinoma, lymphoma, sebaceous carcinoma, and lung adenocarcinoma. Most cases received conventional therapy before being enrolled, with only three naïve patients. Four of the fifteen measurable cases of OMM with stage IV achieved a partial response (26.7%). Moreover, 4 of the 24 melanomas had stable disease (16.7%), two of which were stage III and IV cutaneous melanoma. Nevertheless, the study mentioned that incorporating concurrent continuous treatments such as the tyrosine kinase inhibitor toceranib, metronomic cyclophosphamide, or non-steroidal drugs was permitted if previous treatments had failed to elicit discernible antitumour effects, which may have influenced the stable disease rate.

The author also noticed variation in response to the same tumour in the same individual in different locations (primary versus metastasis and between different metastases). Similar to pseudoprogression (increase in tumour size due to tumour inflammatory infiltrate rather than real progression), mixed tumour responses have also been identified in human studies and highlight the difficulty in assessing response with immunotherapy [81]. The study of Igase et al. (2020) also suggested an improved survival time for the evaluated 16 stage IV oral melanoma patients treated with anti-PD-1 monoclonal antibodies of 166 days (95% CI 56–307) compared to 55 days (95% CI 27–143) of a historical control group of dogs treated at the same institution with various conventional therapies (surgery, radiation therapy, chemotherapy, and/or DNA vaccine) (*p* = 0.046) [79]. However, other studies have shown a possible longer median survival time of 80–90 days for stage IV melanoma with various treatments [82,83]. Treatment-related adverse events (AEs) occurred in 63.3% of cases. Most AEs were mild, with only two greater than grade 3, of which one dog developed fever and one patient died of suspected treatment-related immune-mediated pneumonitis.

Interestingly, the same author [84] two years later published a follow-up report on two of the oral melanoma cases with pulmonary metastasis that were classified at the time of writing as partial remission (PR) and progressive disease (PD). The author reported that one of the dogs was previously classified as having PD based on the increasing size of the pulmonary metastases (day 267, after four cycles), but while continuing the treatment, the metastases regressed by day 352 and achieved a complete response, surviving for longer than 2 years (day 954). In the other case that was classified as PR (on day 67), the treatment was continued, and the pulmonary metastases reduced in size. However, as the dog developed a suspected multiple myeloma, the owner elected to stop the treatment. The dog died on day 490, and at necropsy, melanoma was not detected either macroscopically or microscopically, while multiple myeloma was confirmed. All these findings suggest that delayed response rate and/or pseudoprogression, as occur in people, are also possible in dogs.

Just one year after the study of Igase et al., Maekawa et al. [4] investigated the efficacy of the previously used c4G12 in a 2017 pilot study in a larger cohort of 29 dogs with oral melanoma and concurrent pulmonary metastases. Immunohistochemical expression of PD-L1 was also investigated and correlated with response and survival. In this study, 16 dogs (62%) had PD-L1-positive cancers with TPS of ≥50%, one dog had a low score between 1 and 49%, two dogs had PD-L1-negative cancers (TPS < 1%), and seven dogs (24%) were undetermined. Most dogs enrolled received at least one prior treatment, including surgery (17 patients), radiation (14 patients), and/or chemotherapy (1 patient). In 13 dogs with measurable disease, the response rate could be evaluated only in 10 patients. Of these 10 patients, only one (7.7%) achieved a complete response, while all the rest achieved only progressive disease. In this responder dog, both the recurrent primary tumour and the pulmonary metastases were not detectable by week 7 (3rd injection). Interestingly, four of the dogs with non-measurable diseases achieved some form of response. One that received the c4G12 developed metastases at 4 weeks, but the metastases completely responded by week 12. Another dog had one metastatic lesion in the lung that disappeared at week 6, but another lesion showed a slight increase at the same time, followed by complete regression at week 18. In total, five patients had some form of response, four were non-measurable at the start of the treatment, and one had measurable disease (17%), of which three achieved CR. The survival of two of the dogs that achieved complete response (CR) had a long-term survival of 417 days and more than 530 days. The survival of all the treated patients was compared to a historical control of dogs treated at the same institution. Survival in the treatment group was statistically significantly longer (*p* = 0.00006), with a median survival of 143 days versus 54 days in the control group. Of the variables investigated as predictive of response, the use of radiation therapy (*p* = 0.02), baseline low plasma CRP (cutoff 2.55, *p* = 0.01), and high lymphocyte/monocyte ratio (LMR) (cutoff 1.41, *p* = 0.0002) were statistically significant. The association between tumour PD-L1 expression and clinical outcome was unclear, as the majority of dogs had PD-L1-positive cancers with a TPS ≥ 50%. However, of all the five dogs that achieved a response, one had PD-L1 TPS of <1%. The authors concluded that in dogs, PD-L1 expression could not be considered a good marker of response, but larger studies are needed. Similarly to other studies, the adverse events were mild, with around 50% of AEs, only 13% of dogs in grade 3, and no dogs dying of treatment. Suspected immune-related causes of pneumonitis and thrombocytopenia were also reported in one and two dogs, and other cases of suspected immune-mediated hepatitis, pancreatitis, and colitis were suspected but not clinically confirmed.

From all the short studies performed, it remains unclear why only a few dogs with melanoma respond to PD-1/PD-L1 monoclonal antibodies and if any markers could be predictive of response to PD-1/PDL-1 blockade. In an attempt to answer these questions, Makaewa et al. (2022) investigated the serum concentrations of IFN-γ, IL-2, IL-6, IL-10, IL-12/IL-23p40, TNF-α, IL-8, MCP-1, NGF-β, SCF, and VEGF-A in dogs with oral melanoma with pulmonary metastases treated with the canine chimeric anti-PD-L1 antibody c4G12 [85]. The author first compared serum concentrations between healthy controls (*n* = 8) and dogs with pulmonary metastatic OMM (*n* = 27). PGE2, IL-12p40, IL-8, MCP-1, and SCF were found significantly in higher concentrations in OMM dogs than in the healthy control dogs. Amongst the dogs that received c4G12, higher serum concentrations of PGE2, MCP-1, and VEGF-A were associated with worse OS (*p* = 0.038), while higher concentrations of IL-2, IL-12p40, and SCF were correlated with improved OS (*p* = 0.045). Low serum levels of PGE2 (<6.5) and IL-6 (<45.8) were significant predictors of longer survival in dogs with anti-PD-L1 therapy (*p* = 0.033 and 0.031, respectively), whereas IL-2 and IL-6 predicted tumour response to c4G12 treatment (*p* = 0.030 and 0.013, respectively). ROC analysis was performed for each serum cytokine to assess the correlation between the patients‘ longer survival and tumour response and showed that PGE2 and IL-6 were significant predictors of longer survival (*p* = 0.033 and 0.031, respectively), while IL-2 and IL-6 predicted tumour response (*p* = 0.030 and 0.013, respectively). The authors speculate that the COX-2/PGE2 pathway could suppress antitumour immunity, conferring resistance to c4G12. Despite this, the authors did not investigate COX-2 tumour expression, while to prove the immunosuppressive property of serum PGE2 in dogs, they performed an in vitro experiment. Peripheral blood mononuclear cells (PBMCs) were collected from healthy dogs and cultured with non-specific T cell stimulators in the presence of prostaglandin E2 (PGE2). PGE2 treatment significantly reduced IL-2 and IFN-γ production by stimulated canine PBMCs, and the author concluded that PGE2 could be a potent suppressor of canine T-cell responses. Considering the widely used COX-2 inhibitor meloxicam, the author investigated with a similar in vitro experiment the effect of combining meloxicam and an anti-PD-L1 antibody and showed enhanced IL-2 production and significantly increased IFN-γ production, suggesting that dual blockade of the PD-1/PD-L1 and COX-2/PGE2 pathways could enhance antitumour immune responses in dogs, at least in vitro.

To investigate possible ways to boost immune response to ICI treatment, Deguchi et al. (2023) retrospectively investigated if radiotherapy could increase response to the previously investigated c4G12 anti-PDL1 monoclonal antibody treatment. The author included 39 dogs diagnosed with pulmonary metastatic OMM and treated with c4G12 and divided them into three groups: i.e., one group of twenty dogs that did not receive radiotherapy or received radiotherapy more than 8 weeks from the start of c4G12 treatment (no RT); the second group of nine dogs that received previous radiotherapy within 8 weeks from the start of c4G12 (previous RT); and the third group of ten dogs that were treated concurrently within one week of c4G12 (concurrent RT) [80]. All dogs were treated with hypofractionated radiotherapy for the primary oral melanoma, and the three groups were considered reasonably homogeneous. The response rate and the clinical benefit (CR + PR + SD) of the pulmonary metastases were higher for the previous RT group (4 CR, 1 SD/PR) compared to the no RT group (1 CR, 1 SD/PR) and concurrent RT group (0 CR, 2 PR/SD), and this was also not statistically significant. The median OS from the first c4G12 treatment to death in all 39 dogs was 129 days. The MST in the no RT group, previous RT, and concurrent RT were 185 days, 283.5 days, and 129 days, respectively, with only the previous radiotherapy group being significantly longer versus the no RT group (*p* = 0.036). The author concluded that the systemic antitumour immunity of anti-PD-L1 therapy was enhanced by RT, but the timing of RT is important for achieving synergistic efficacy. However, a few limitations were present in this study, including the small number of patients in each group and the inclusion of dogs in the no-RT group for dogs that still received radiotherapy, even if they were longer than 8 weeks apart. From the experience of immunotherapy treatment in both dogs and humans, it is clear that immune responses can take an unpredictable longer time to manifest, and comparing larger groups of radiotherapy-treated and not-treated patients is necessary to prove these findings. Despite these limitations, the possibility that radiotherapy-induced immunostimulation by abscopal effect is an interesting concept. The abscopal effects have been reported in people [86], and it is reasonable to think that could be a boost for concurrent immunotherapy treatment. The mechanism behind the abscopal effect is not completely clear, but it is potentially induced by the accumulation of excessive cyclic GMP-AMP (cGAMP) derived from circulating tumour DNA. This process stimulates APCs and subsequently promotes T-cell-mediated killing of tumour cells [87]. This phenomenon has been observed after radiotherapy in conjunction with ICIs (CTLA-4 and PD-1/PD-L1 inhibitors), where the efficacy of these immune modulators may be enhanced by the addition of radiotherapy [88].

In 2024, Igase et al. repeated the trial performed in 2020 in a larger cohort of dogs, using only the anti-canine PD-1 monoclonal antibody (ca-4F12-E6) and evaluating the safety and efficacy in dogs with advanced solid tumours except for OMM [89]. Thirty-seven dogs were evaluated for adverse events, and thirty dogs could be assessed for treatment response. This author also performed IHC for infiltrating lymphocytes in one of the cases that achieved CR. The tumour types included five urothelial carcinomas, six SCCs, five undifferentiated sarcomas, three nasal adenocarcinomas, three mammary gland tumours, two of each lung carcinoma, heart base tumours, osteosarcoma, soft tissue sarcoma, skin melanoma, hepatocellular carcinoma, colon-rectal carcinoma, and one of each adenocarcinoma of unknown origin and colon-rectal carcinoma. Most cases achieved various previous treatments. Of the 29 cases for which responses could be evaluated, 1 dog achieved CR (nasal SCC), 1 dog achieved PR (maxillary osteosarcoma), 6 had SD, and 14 had PD. The overall response and clinical benefit rates were 6.9% and 27.6%, respectively [86]. Interestingly, one pulmonary carcinoma and undifferentiated sarcoma showed pseudoprogression and a mixed or discordant response, as previously reported, highlighting again the difficulty in assessing response to immunotherapy [89]. Adverse events were recorded in 13 out of 37 cases (35.1%), with diarrhoea and elevated liver enzymes being the most frequent. AEs of grade 4 were only seen in elevated alanine transaminase, and interestingly, two secondary tumours were also reported as possibly caused by the drugs. Two dogs developed suspected immune-mediated sterile nodular panniculitis, and one developed concurrent myasthenia gravis, megaesophagus, and hypothyroidism. However, overall, the treatment was well tolerated.

Recently, a caninized monoclonal antibody against PD-1 has been conditionally licensed in the USA for the treatment of dogs with mast cell tumours and melanoma gilvetmab^®^ (Merck & Co., Inc., Rahway, NJ, USA). This new drug resulted in 73% and 60% clinical benefits in mast cell tumours and melanoma, respectively [90]. However, in the same study, the objective response rate (CR + PR) in the oral melanoma group was only 20%. Despite the preliminary results, further studies are needed to confirm the real benefits of ICI in dogs. Considering the cost and challenge of producing and making available in the small oncology veterinary market caninized monoclonal antibodies, an in vitro study was conducted to investigate the potential human-dog cross-reactivity of approved human ICI [91].

Seven FDA-approved human ICIs against the main targets CTLA-4 and PD-1/PD-L1 were tested in dogs in vitro. The author used flow cytometry and ELISA to evaluate cross-reactivity and blocking capacity while evaluating functional responses and cytokine formation in peripheral blood mononuclear cells (PBMCs) obtained from dogs with cancer (*n* = 27) and healthy donors (*n* = 12). The study found that four out of seven human ICI monoclonal antibodies were cross-reactive, and especially atezolizumab and avelumab showed blocking activity of the canine PD1/PD-L1 axis. Atezolizumab also achieved significant functional benefits when used in cPBMC activation in vitro assays, increasing cIFN-γ production in both samples from healthy and cancer patients and showing a similar binding profile to canine PD-L1 when compared to human PD-L1. The author concluded that this in vitro study provides the rationale for clinical testing of atezolizumab in dogs. Despite the author acknowledging that being an in vitro study, adverse events or the development of anti-drug antibodies could not be predicted, it also argues that studies investigating the pharmacokinetic parameters of other humanised monoclonal antibodies have already been performed and did not show significant anti-drug antibodies that resulted in noticeable decreases in drug concentration, and neither severe adverse events nor severe adverse events were reported. This could also be the case for atezolizumab; the author finally speculates that canine Fc gamma receptors (FcγRs) and neonatal Fc receptors (FcRn) could be compatible between humans and dogs [92]. Table 3 presents the above-discussed studies on the clinical and treatment data of canine OMM to specific PD-1/PD-L1 ICIs.

## 4. Conclusions

Overall, the response rate of caninized antibodies in oral melanoma in dogs, at least at this stage, is a bit disappointing, with around or less than a 20% response rate compared to a 40–60% response rate in cutaneous melanoma in people. However, it is possible that oral melanoma is not the most promising target of canine ICI. Evaluating factors such as PD-L1 tumour expression, tumour-infiltrating lymphocytes, and molecular/genetic signatures could help predict response to ICI in certain types of dog tumours. The selection of the patient that benefits the most from potentially expensive ICI must focus on finding markers predictive of the response and potentially screening individual tumours rather than only considering a specific type or class of tumours. The improvement of the prediction of response to target therapy would contribute to the development of personalised medicine in dogs.

## Figures and Tables

**Figure 1 cancers-16-02003-f001:**
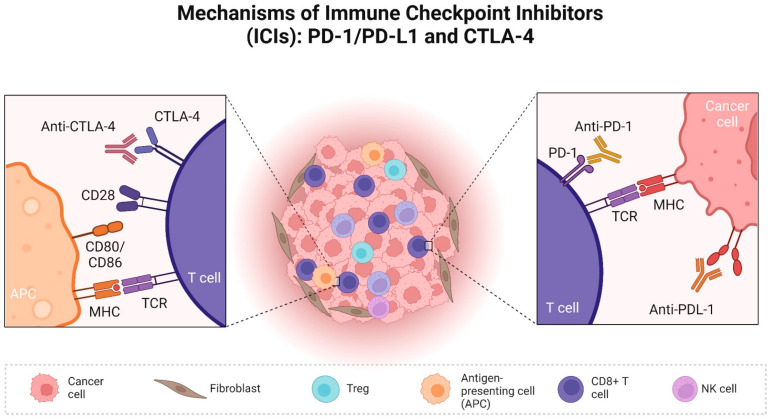
Mechanisms of blockage of CTLA-4 and PD-1/PD-L1 immune checkpoints within the tumour microenvironment. CTLA-4 blockage is illustrated by the interaction between CTLA-4 on a T cell and an anti-CTLA-4 ICI. This interaction prevents the inhibitory signals mediated by CTLA-4, while its ligands CD80/CD86 on an antigen-presenting cell may bind to CD20 instead of CTLA-4 on the T cell. PD-1/PD-L1 blockage is illustrated by the binding of either PD-1 on a T cell to an anti-PD-1 ICI or PD-L1 on a cancer cell to an anti-PD-L1 ICI. In both cases, T-cell receptor binding to MHC is facilitated, enabling proper antigen recognition within the tumour microenvironment. The comprehensive modulation of the immune system within the tumour microenvironment involves various components, including T regulatory cells, B cells, NK cells, and cytotoxic T cells (CD4^+^). Additionally, the tumour cells can shape the stroma by influencing fibroblasts as well as the vascular and lymphatic endothelium.

**Table 1 cancers-16-02003-t001:** PD-L1 expression in different cancers in dogs according to Maekawa et al. (2021) [4] and Muscatello et al. (2023) [59].

Pathology	Maekawa et al. (2021) [4]		Muscatello et al. (2023) [59]	
	Positive Immunolabeling	TPS (≥50%)	Positive Immunolabeling	TPS
Melanoma	19/20 (95%)	18/20 (90%)	17/17 (100%)	1–14%
Squamous cell carcinoma	18/20 (90%)	16/20 (80%)	3/17 (18%)	17–64%
Mammary gland carcinoma	20/20 (100%)	20/20 (100%)	1/31 (3%)	9%
Apocrine gland anal sac adenocarcinoma	19/20 (95%)	15/20 (75%)	-	-
Nasal adenocarcinoma	20/20 (100%)	18/20 (90%)	-	-
Pulmonary carcinoma	-	-	2/20 (10%)	78–97%
Renal carcinoma	-	-	4/17 (24%)	16–91%
Urothelial carcinoma	20/20 (100%)	20/20 (100%)	2/18 (11%)	29–69%
Gastric carcinoma	4/5 (80%)	3/5 (60%)	0/10	-
Intestinal carcinoma	-	-	0/24	-
Lymphoma	17/20 (85%)	12/20 (60%)	2/14 (31%)	1–13%
Transmissible venereal tumour	0/4 (0%)	-	-	-
Soft-tissue sarcoma	14/20 (70%)	12/20 (60%)	-	-
Histiocytic sarcoma	18/20 (90%)	6/20 (30%)	-	-
Osteosarcoma	20/20 (100%)	16/20 (80%)	-	-

TPS: Tumour proportion score (percentage of positive neoplastic cells).

**Table 2 cancers-16-02003-t002:** CTLA-4 expression in different samples of tumours in dogs according to Tagawa et al. (2018), Ariyarathna et al. (2020), and Porcellato et al. (2021) [68,69,72].

Sample	B Cell High Grade Lymphoma	Mammary Gland Tumour	Melanoma
Tumour cells	Unknown	IHC	IHC and RT-qPCR
Serum	Unknown	Unknown	Unknown
TIL CD4^+^	Flow cytometry	Unknown	Unknown
TIL CD8^+^	Flow cytometry	Unknown	Unknown
Serum CD4^+^	Flow cytometry	Unknown	Unknown
Serum CD8^+^	Flow cytometry	Unknown	Unknown
References	Tagawa et al., 2018 [68]	Ariyarathna et al., 2020 [69]	Porcellato et al., 2021 [72]

**Table 3 cancers-16-02003-t003:** Summary of ICI studies in oral malignant melanoma.

Study	ICI Agent	Number of Dogs	Response Rate	Median Survival Time	Clinical Staging (WHO)
Deguchi et al. (2023) [80]	Chimeric rat–dog anti-PD-L1	39 (TG)	CR in 5/34, PR/SD in 4/34 *, and PD in 25/34	129 days (TG) and 48 days (HCG)	Stage IV (all with pulmonary metastases)
Maekawa et al. (2021) [4]	Chimeric rat–dog anti-PD-L1	29 (TG) and 15 (HCG)	CR in 1/10 and PD in 9/10	143 days (TG) and 54 (HCG)	Stage IV (all with pulmonary metastases)
Igase et al. (2020) [79]	Chimeric rat–dog anti-PD-1 and caninized anti-PD-1	21 (TG, 15 in the chimeric ICI group and 6 in the caninized ICI group) and 23 (HCG)	PR in 4/15, SD in 1/15, and PD in 10/15	166 days (TG) and 55 days (HCG)	Stage III (4) and stage IV (17)
Maekawa et al. (2017) [5]	Chimeric rat–dog anti-PD-L1	7 (TG) and 17 (HCG)	PR in 1/7 and PD in 6/7	93.5 days (TG) and 54 days (HCG)	Stage II (1), stage II (2), and stage IV (4 with pulmonary metastases)

CR: complete response; HCG: historical control group; PD: progressive disease; PR: partial response; TG: treatment group. * The study did not differentiate the patients in PR and SD.

## Data Availability

The raw data supporting the conclusions of this article will be made available by the authors on request.

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
