# Peer review of "Checkpoint Inhibitors in Dogs: Are We There Yet?"

_cancers, 2024, doi:10.3390/cancers16112003_

Round 1

Reviewer 1 Report

Comments and Suggestions for Authors

 1) Brief Summary: Checkpoint inhibitors in pet dogs. Are we there yet?

            According to the authors, " This article aims to review the current research literature about the expression of immune checkpoints in canine cancer and the current results of ICI treatment in dogs."

            The text was divided into three sections: 1) Introduction, 2) The expression and clinical relevance of immune checkpoints in dogs with cancer, and 3) Conclusions.

2) General concept comments

            This review is very interesting and informative. However, a thorough review of style and addressing critical issues is required to improve the merit of this work.

            My suggestions for many minor style changes are found in the attached file. The main issues are discussed below.

I. The Title. This review focused on domestic dogs, not wild ones. Nevertheless, the expression "pet dogs" appears redundant; I suggest removing the word "pet."

II. The Simple Summary and the Abstract are adequate.

III. The Introduction is too long. The authors made the decision to review the general information regarding checkpoint proteins PD-1/PDL1, CTLA-4, LAG-3, TIM-3, CD200 63, and OX-40. The text is like the review itself, not an Introduction. Even the objective is missing (present only in the summary and abstract). My suggestion is to reduce the Introduction and start the second chapter, "The expression and clinical relevance of immune checkpoints in dogs with cancer", with a section called "Role of Check Point Proteins".

IV. The expression and clinical relevance of immune checkpoints in dogs with cancer

            There are issues and suggestions regarding this section that are presented below.

1) I see no reason for the authors to inform the value of p obtained by the cited reports. Published results are supposed to be reviewed and trustworthy unless they are making a very specific point; the frequent citation of the p-value is irrelevant and truncates the text unnecessarily. Please remove the p-values.

2) The paragraphs are too long in general. This is particularly so in section 2.3, "Clinical trials of Immune checkpoint inhibitors in dogs." Long paragraphs are harder to read and grasp the meaning intended by the authors. Please use one main idea per paragraph only.

3) The sentence "Although comparison within trials is not possible, there was a comparable efficacy..." (lanes 118-119) is a contradiction in itself. If comparison is not possible, efficiency can not be comparable.

4) Table 1 should be renamed "Summary of the published literature of PD-L1 expression in different cancers in dogs." It contains information from two papers. I suggest "PD-L1 expression in different cancers in dogs according to Maekawa et al. (2021) and (4) Muscatello et al. (2023) (59)". I also suggest that the columns related to the results of Maekawa et al. are presented on the left, following alphabetical and chronological criteria.

5) I also suggest a change in the presentation of Figure 1. The PD-1 checkpoint is presented first in the text, followed by CTLA-4. In the Figure, CTLA-4 comes first (on the left side). CTLA-4 should come on the right side, and the caption should read " Mechanisms of blockage of PD-1/PD-L1 and CTLA-4 immune checkpoints within the tumour microenvironment.

6) I have made many suggestions regarding the style of the English language. The suggestions are in the attached file.

V. The Conclusions

            The conclusion is not appropriate and should be rewritten. My suggestions are:

1) The sentences below are characterized as discussion, not as conclusions, and should be stricken:

Overall, the response rate of caninized antibodies in oral melanoma in dogs, at least at this stage and from the Japanese experience, is a bit disappointing with around or less than a 20% response rate compared to a 40-60% response rate in cutaneous melanoma in people. However, it is possible that oral melanoma is not the most promising target of canine ICI. As previously discussed, it has been proven that human cancers with microsatellite instability and/or high tumour mutational burden (TMB) are more likely to respond to ICI. While cutaneous melanoma in people has a high TMB, this is not true for oral melanoma in dogs. Other tumours with a high TMB in dogs may have a better response and occasionally marked response has been seen in different other tumours, as reported in the previously described trial of ICI in dogs.

2) My suggestion for the conclusion is:

  Evaluating factors such as PD-L1 tumour expression, tumour-infiltrating lymphocytes, and molecular/genetic signatures could help predict response to ICI in certain types of dog tumours. The selection of the patient that benefits most from potentially expensive ICI must focus on finding markers predictive of the response and potentially screening of individual tumors rather than considering a specific type or class of tumors. Improving personalized medicine for dogs could help improve response prediction.

Comments on the Quality of English Language

Please refer to attached files.

Author Response

File attached

Reviewer 2 Report

Comments and Suggestions for Authors

This is a pertinent and up-to-date article that provides a concise review of the literature on the subject, highlighting the necessary path for veterinary medicine to follow in the search for more scientific evidence on oncology and prognostic factors. I would only suggest changing a few details:

- Sample Summary: Same sentences repeated in simple summary and abstract as: Among ICIs, 35 anti-PD-1 and anti-PD-L1 treatments stand out as the most promising, mirroring the success in human medicine over the past decade.

- line 34: "on these species" instead of " in dogs" at the end of the sentence

- line 97: Use only "mAb"

- line 154: leukemia

- line 320-321: What is the reference of this grading?

- line 323: Put in vitro in italic

- line 329: indicated the meaning (oral malignant melanoma)

- line 345: put the year of publication

- line 357: PR and PD - indicate the meaning

- line 367: name of the author is missing - Maekawa et al.

- line 370: 16

- line 396-397: Put (AE) and revise and improve the meaning of the sentence

- line 422: in vitro in italic and explain the sentence -  peripheral blood mononuclear cell

- line 423: Also write Prostagladin E2

- line 427: in vitro in italic

- line 470: also write complete remission

- line 476: write partial remission

Author Response

file attached

Reviewer 3 Report

Comments and Suggestions for Authors

In the manuscript the authors performed a short introduction on the immune checkpoint proteins and their blockage in human oncology and reviewed the knowledge of this topic in veterinary oncology (in dogs). Despite this a hot topic in oncology, the development of immunotherapy in dogs is very incipient and I not sure that this is the moment for a review on this topic. This could confirm by the fact that only two studies of the PD1 and PDL1 were included in the table 1 and for CTLA4 only 3 studies described the expression of this protein in canine tumors.  Actually, a table with the data of the CTLA4 studies (similar to table 1) is missing and should be included. Similarly, the data of the few clinical trials of Immune checkpoint inhibitors in dogs should be represented schematically.

Minor points

Simple summary: ICI should be fully written before it been used.

Line 31: “caninized antibodies”. I am not sure that caninized exist?

Line 232-233: the study of PD1/PDL1 and CTLA4 in canine mammary gland used the intensity of the immunolabelling for scoring. What is the opinion of the authors about this? The authors should provide a critical revision of the different scoring systems used in veterinary oncology compare to those use in human (and that are validated). This discussion would certainly increase the value of the manuscript and the utility of this review for the future studies on this topic in veterinary oncology.

Line 285: CTVT should be fully written

Figure 1: the legend is very short. Please provide a full explanation so that the figure and the legend could be interpreted without the text in the manuscript.  

Line 312: what was “objective response”?

Line 329: OMM should be fully written in the first use in the manuscript.

Author Response

file attached

Reviewer 4 Report

Comments and Suggestions for Authors

This review is interesting and well-written. It comprehensively presents the state-of-the-art knowledge about clinical trials of human and canine tumor immunotherapy with special emphasis being put on canine tumors. Of particular interest is a comparative synthesis of immunotherapy efficacity in human and canine cancers and factors/predictors affecting it. 

Minor issues:

line 81 'mAB' should be replaced by 'mAb'

line 97 'mAb' term should be used and not  'monoclonal antibody'

line 285 'CTVT' abbreviation should be explained

line 287 'TVT' abbreviation should be explained

line 329 'OMM' abbreviation should be explained while used for the first time

line 410 in the sentence '...higher serum concentrations of PGE2.......were associated with worse OS.'  but in the following sentence (line 412) there is an opposite statement: 'PGE2....were significant predictors of longer survival'. Authors should rephrase the latter sentence to better explain that low-level PGE2 is a predictor of longer survival.

Author Response

file attached

Round 2

Reviewer 3 Report

Comments and Suggestions for Authors

The revised version of the manuscript is improved. The introduction of the tables really contributed to this improvement.

Minor points should be corrected:

1) abstract line27: immune is in capital letters. Please correct

2 the subheading title should be reformulated: 2. Role of Check Point Proteins in human cancer and target therapy

3) lines 245-24. Please rephrased. I am providing a suggestion:  However, the immunohistochemistry score systems for evaluating staining intensity could lack reproducibility, be subject to pathologist interpretation and to difference in the time/type of chromogen used to visualize the immunoreaction. This jeopardizes the comparison between the results of veterinary studies on this topic.

4) Conclusion. I think the last sentence should be on contrary. The improvement of the prediction of response to target therapy would contribute to develop the personalized medicine in dogs.

Author Response

Thank you for your comment and your help to improve the manuscript. We have revised the manuscript according to your comments and made all the changes in the text.